# Positional Distribution of Fatty Acids in Processed Chinook Salmon Roe Lipids Determined by ^13^C Magnetic Resonance Spectroscopy (NMR)

**DOI:** 10.3390/molecules28010454

**Published:** 2023-01-03

**Authors:** Senni Bunga, Mirja Kaizer Ahmmed, Alan Carne, Alaa El-Din A. Bekhit

**Affiliations:** 1Department of Food Sciences, University of Otago, P.O. Box 56, Dunedin 9054, New Zealand; 2Fishing and Post-Harvest Technology, Chittagong Veterinary and Animal Sciences University, Khulshi, P.O. Box 14, Chittagong 4202, Bangladesh; 3Department of Biochemistry, University of Otago, P.O. Box 56, Dunedin 9054, New Zealand

**Keywords:** positional distribution, omega-3, EPA, DHA, NMR, marine lipid

## Abstract

Recently, there has been great interest in the lipidomic of marine lipids and their potential health benefits. Processing of seafood products can potentially modify the characteristics and composition of lipids. The present study investigated the effect of processing methods (salting and fermentation) on the positional distribution of fatty acids of Chinook salmon roe using ^13^C nuclear magnetic resonance spectroscopy (NMR). The NMR analysis provided information on the carbonyl atom, double bond/olefinic, glycerol backbone, aliphatic group, and chain ending methyl group regions. The obtained data showed that docosahexaenoic acid (DHA) is the main fatty acid esterified at the *sn*-2 position of the triacylglycerides (TAGs), while other fatty acids, such as eicosapentaenoic acid (EPA) and stearidonic acid (SDA), were randomly distributed or preferentially esterified at the *sn*-1 and *sn*-3 positions. Fermentation of salmon roe was found to enrich the level of DHA at the *sn*-2 position of the TAG. The processing of roe by both salt drying and fermentation did not appear to affect the proportion of EPA at the *sn*-2 position. This present study demonstrated that fish roe processing can enhance the proportion of DHA at the *sn*-2 position and potentially improve its bioavailability.

## 1. Introduction

Salmon roe lipid has attracted increasing interest due to its favorable nutritional composition [1,2,3]. In particular, the lipid composition of caviar [4], fermented salmon roe [5], and salted salmon roe [6] have been recently reported. Salmon roe is reported to contain a considerable amount of phospholipid (26.53 µmol/g) and n-3 fatty acids (43.32% of the total fatty acid content) [7]. The omega 3 polyunsaturated fatty acids (n-3 PUFA) esterified as phospholipid present in salmon roe are beneficial for the treatment of cardiovascular heart disease, brain health, chronic liver diseases, and non-alcoholic fatty liver disease [8,9,10].

The composition of fish roe is affected by season, maturity, sex, culture technique, geographical location, and processing method [4]. Different processing regimes can have different effects on the composition of nutrients and bioactive compounds, including fatty acid, amino acid, mineral, vitamin, and antioxidant contents. The stability, quantity, and bioavailability of omega-3 fatty acids depend on the processing methods used in the preparation of the food, or extraction of the lipids [9,10]. In previous studies, we reported the effects of salting [6] and fermentation [5] on the fatty acid and phospholipid compositions and contents during the preparation of salmon karasumi and karashi mentaiko-like products, respectively. The effects of these processing methods on the positional distribution of n-3 fatty acids were not reported, but such information can be very useful to understanding changes reported in these previous studies, especially those relevant to the n-3 PUFA.

High-resolution NMR techniques, such as ^1^H NMR, ^13^C NMR, and ^31^P NMR have been used to provide valuable information at the molecular level of complex lipid systems using a non-destructive approach. Direct and comprehensive observations for the detection and quantification of fatty acid composition, phospholipid composition, metabolomic information, and positional distribution of particular fatty acids can also be made with minimal sample preparation using NMR [7,11,12,13]. The application of the ^13^C NMR technique has been found to be very useful for obtaining additional structural information, such as the determination of positional distribution on the glycerol backbone of different classes of fatty acids, and observation of carbonyl signals [12]. The regional specificity characteristics of n-3 PUFAs in the TAGs when linked to α and β chains, which represent the *sn*-1,3 and *sn*-2 positions, respectively, can also be identified by ^13^C NMR [12]. Moreover, the positional distribution of n-3 PUFA that have fatty acids with double bonds close to the carboxyl group in fish oils can be assessed quantitatively by ^13^C NMR analysis [12]. ^13^C NMR analysis has been reported for the determination of the positional distribution of fatty acids on the glycerol backbone in hoki and tuna oils [14], salmon roe [7], and other fish samples [15,16,17]. ^13^C NMR has also been applied effectively in combination with chemometric methods to authenticate oils from different fish species [18], to differentiate between wild and farmed Atlantic salmon [19], and to analyze fish oil capsules produced from oil of different fish species [20]. In a previous study, we reported the positional distribution of n-3 fatty acids of fresh King salmon roe using ^13^C NMR analysis [7]. However, the study was only focused on the carbonyl region and quantified only n-3 fatty acids (EPA; docosapentaenoic acid, DPA; and DHA). Knowledge of the positional distribution of fatty acids in salted and fermented roe has not been reported.

Therefore, the present study aimed to investigate the regional specificity of fatty acids in lipids present in salt dried and in fermented Chinook salmon roe products using ^13^C NMR spectroscopy.

## 2. Results

### 2.1. ^13^C NMR Analysis of Salmon Roe Extracted Lipids

Lipids were extracted from fresh (FR), salted dried (SD) and fermented (SF) Chinook salmon roes, based on total lipid extraction established previously [21]. Samples then were subjected to ^13^C NMR analysis as described earlier. ^13^C NMR signals can be grouped into five main regions corresponding to different functional groups [18,20] that include carbonyl atom, double bond/olefinic, glycerol backbone, aliphatic group, and chain ending methyl group regions (Appendix A). 

#### 2.1.1. ^13^C NMR Carbonyl Region (174.3–172.6 ppm)

The positional distribution of major n-3 fatty acids (EPA and DHA), total saturated fatty acids (SFA), and total mono-unsaturated fatty acids, in fresh, fermented, and dried salted salmon roe were quantified using ^13^C NMR and the results are shown in Table 1. The carbonyl signal region can be used to identify the positional distribution of major n-3 PUFA in positions *sn*-2 and *sn*-1,3 of triacylglycerides (TAG) [22]. Any chemical modification or adulteration of the natural oils can be detected in this region [20]. The major n-3 fatty acids in this region are C22: 6n-3 (DHA), C2: 5n-3 (EPA), and C18: -3 (SDA), where the characteristic peaks from the main n-3 fatty acids in *sn*-1,3 and *sn*-2 positions of the TAG are identified for all samples in the chemical shift 172.6–174.3 ppm region [23]. The signals for DHA in both fresh and processed Chinook salmon roe lipids were found to be preferentially located at the *sn*-2 position and had a higher percentage area compared to the signals for EPA and SDA at the same position (Table 1). EPA and SDA were preferentially located at *sn*-1 and *sn*-3 compared to DHA (Table 1). Fermentation of salmon roe increased the percentage of DHA at the *sn*-2 position from 6.84% (FR) to 7.11% (SD) and 10.09% (SF), and decreased the percentage at the *sn*-1,3 position from 0.99% (FR) to 0.80% (SD) and 0.68% (SF) (*p* < 0.05). This suggests that fermentation might facilitate complex biochemical reactions, which substantially alter the chemical structure of the lipid due to the activation of various lipases, that could result in hydrolysis and re-esterification of fatty acids on triacylglycerides [24,25]. This could be the reason for the higher DHA content at the *sn*-2 position compared to the *sn*-1,3 position of fatty acids in extracted lipid following fermentation of salmon roe samples. DHA esterified at the *sn*-2 position is reported to be highly bioavailable and able to cross the blood/brain barrier [26,27], suggesting that consumption of fermented salmon roe could facilitate brain and mental health. 

Conversely, the area % of EPA at the *sn*-2 position in the lipid of fresh salmon roe was found to be higher (*p* < 0.05) than in the dried salted and fermented roe lipid (Table 1). Stereospecific studies on fish oils, such as sardine [28], menhaden [29], tuna [30], and bonito head [31] have reported that EPA was preferentially located at the *sn*-3 position, followed by the *sn*-2 and *sn*-1 positions, which is consistent with the present study and can explain the higher level of EPA at the *sn*-1,3 position compared to the *sn*-2 position. 

The positional distribution of SFA and MUFA in salmon roe in the present study was found to be preferentially located at the *sn*-1,3 position, both before and after processing, corresponding to the higher area percentage found at the *sn*-1,3 position. Studies of hoki and tuna oil using ^13^C NMR have reported that DHA is esterified more at the *sn*-2 than at the *sn*-1,3 position [14]. A higher DHA content at the *sn*-2 position compared to *sn*-1,3 position is also consistent with a previous study on fresh salmon roe [7]. However, from reports in the literature, the positional specificity of EPA appears to vary depending on the fish species. For example, fish oils from mackerel and herring [32], and saury, herring, and capelin [33], found that EPA was preferentially located at the *sn*-2 position, followed by the *sn*-3 and *sn*-1 positions, which are the same as for olive oil triacylglycerides [25]. Previous literature [34] reported that in most commercial fish oils, EPA has no particular positional distribution, whereas DHA is more usually esterified at the *sn*-2 position in TAG molecules. Both salting and fermentation have been reported to activate various indigenous lipases [35], that could alter the positional distribution of SFA and MUFA esterified in lipids, resulting in the observed change in MUFA distribution as a result of various treatments. Interestingly, both the salted and the fermented roe samples were found to have a lower SFA content in the *sn*-1,3 position compared to fresh roe. However, the mechanism behind this difference is still unclear and could be addressed in future research.

#### 2.1.2. ^13^C NMR of Olefinic Region (132.5–126.5 ppm) 

According to Aursand et al. [20], differences in the fatty acid composition among species, in relation to geographical origin, can be explored using this ^13^C NMR region. Olefinic carbon atoms in n-3 fatty acids show characteristic peaks due to the influence on the chemical shift from any methyl end group in the surrounding area. Gunstone et al. [36] reported that the olefinic carbon resonances are strongly dependent on the distance of the carbons from carboxyl and methyl ends, in addition to the stereochemistry and number of double bonds in the molecular chain. The two resonances, C3, n-3 and C4, n-3 at 132.0 and 127.0 ppm, respectively, are unique for the n-3 fatty acids with a double bond at the C3 position in relation to the methyl end. These two peaks facilitate quantification of the relative concentration of n-3 fatty acids in lipid mixtures [37]. 

In this study, the lipids extracted from dried salmon roe (SD) had a higher concentration (20.03%) of long-chain polyunsaturated fatty acids (22:6n-3, 22:5n-3, 20:5n-3, 20:4n-3, and 18:4n-3) than the lipids extracted from fresh (FR) (17.24%) and fermented (SF) (9.20%) salmon roe, as found at 128.03 ppm (Table 2). Furthermore, the spectral signals of monounsaturated fatty acids (16:1, 18:1, 20:1, and 22:1), observed with the resonance at 129.66–129.96 ppm, revealed that lipids extracted from salted salmon roe had a higher (*p* < 0.05) 16:1, 18:1, 20:1, and 22:1 content than lipids extracted from fresh and fermented salmon roes. The carbon peak assignment for C5, 20:5n-3; 20:4n6 at 128.78 ppm and C5, 22:6n-3 at 129.42 ppm, showed that the integrated peak areas of SF samples were almost 15 times higher than those found in FR and SD samples. Other carbon signals (C6, 20:5n-3; 20:4n6) observed in this region at 129.52 ppm showed that fresh roe (FR) samples had two to three times higher (*p* < 0.05) area peaks than that found in SD and SF roe samples. The C4 carbon in 22:6n-3 fatty acid at δ = 127.59 ppm proportion decreased when salmon roe was fermented. According to the results (Table 2), the concentration of C4, 22:6n-3 of SF roe lipid samples was 2.52%, compared to 4.13% in FR roe lipid, but the proportion increased to 7.7% in the SD lipid. Interestingly, a peak located at 130.84 ppm was found in SF salmon roe lipid that was not detected in either the FR or SD roe lipid, but had a proportion of 17.94% in the SF salmon roe lipid (Table 2). Alexandri et al. [24] reported detecting a signal at δ = 130. 87 ppm which was identified as β-eleostearic acid. Therefore, the peak found in the present study may be representative of the chemical shift of β-eleostearic acid (*trans*-9, *trans*-11, *trans*-13 18:3). β-eleostearic acid has been reported as a conjugated linoleic acid (CLA), a beneficial functional lipid produced by lactic acid bacteria that has the ability to reduce carcinogenesis, atherosclerosis, and body fat [38].

#### 2.1.3. ^13^C NMR Glycerol Carbon Region (61.0–72.0 ppm)

The chemical shift from 61.0–72.0 ppm (Figure 1) displays an enlarged view of the glycerol carbon region in mono-, di-, and triacylglyceride.

The glycerol carbon data are influenced by the stereospecific conformation and the esterified fatty acids, specifically the distance to the nearest double bond. In the present study, the glycerol carbon region indicates that several glycerol esters are present only in the fermented product, specifically 1MAG *sn*-1, 1MAG *sn*-3, and 1,3DAG *sn*-1,3 (Table 3), which were detected at 64.28 ppm, 65.83 ppm, and 64.80 ppm, respectively. The 1MAG *sn*-1 (64.28 ppm) signal contributed 10.21% of the area %, while 1MAG *sn*-3 (65.83 ppm), and 1,3DAG *sn*-1,3 (64.80 ppm) contributed 7.87% and 5.46%, respectively. The glycerol carbon atoms are stereo-specifically numbered by convention as sn-2 (center) and *sn*-1 and *sn*-3 (outer) [39]. According to Baiocchi et al. [40], different combinations of these fatty acids form different TAG molecular species, and the composition of these in each oil/fat is unique, resulting in complex TAG mixtures. In addition, investigation of the stability of fatty acids in regiospecific positions in TAGs has found that fatty acids bonded to the sn-2 position of TAGs are more stable to oxidative degradation and thermal polymerization, compared to those esterified in the *sn*-1 and *sn*-3 positions [41].

#### 2.1.4. ^13^C NMR Aliphatic Region (32.0–20.0 ppm) and Methyl End Chain Region (~14.0 ppm)

The fatty acid assignment of the ^13^C NMR spectrum of Chinook salmon roe lipid extracts obtained from the aliphatic region is presented in Appendix A The chemical shift range of the aliphatic carbon and methyl end chain (Appendix A), although wide (signal at δ = ~14 ppm and δ = ~32 ppm), has been utilized extensively for the identification and quantification of lipids [24]. For the aliphatic region (32.0–31.0 ppm), SF roe lipid appears to have a higher area percentage of SFAs compared with the SD and FR roe lipid samples. The peak from SD roe lipid had higher C3, n-9 compared to FR and SF roe lipid (Table 4, *p* < 0.05). There was no difference (*p* > 0.05) in C3, n-7 content among the three roe samples, although C3, n-6 was higher (*p* > 0.05) in the lipid extracted from the SF roe, followed by the FR roe, and was the lowest in the SD roe. 

## 3. Materials and Methods

### 3.1. Chemicals and Materials

All chemicals and reagents used in the present work were of analytical grade of the highest purity available. Methanol and chloroform were from Merck (Darmstadt, Germany), and deuterated chloroform (CDCl_3_, 99.8%) was obtained from Cambridge Isotope Laboratories, Inc., Tewksbury, MA, USA. Fresh, salted-dried, and salted-fermented salmon roe samples were collected for analysis on 0, 20, and 30 d, respectively, of processing. Fermentation and salting of salmon roe were carried out based on the methods described previously [5,6]. Roe samples then were freeze dried and ground prior to extraction of the lipids. 

### 3.2. Roe Lipid Extraction

Lipids from salmon roes subjected to different individual processes (salt drying and fermenting) were extracted using a methanol–chloroform–water mixture according to the method described previously [21]. The lipid extraction was performed in triplicate for each roe. Samples of 1 g freeze-dried roe material were mixed with 15 mL of chloroform–methanol (2:1, *v/v*) and homogenized using an IKA^®^ T25 Digital Ultra Turrax^®^ (IKA, Staufen, Germany) at 6000 rpm for 2 min. After the addition of 3 mL of Type 1 Milli-Q water, the samples were vortexed, and then centrifuged (Beckman GPR Centrifuge, Fullerton, California, USA) at 2800× *g* for 10 min at 4 °C, and the methanol/water phase was separated from the lipid-containing chloroform phase. The chloroform fraction was dried using a rotary evaporator (Büchi, Re 111, W/B-461 heating bath, Flawil, Switzerland) at a temperature of 30 °C, and the lipid fraction was used for the NMR analysis.

### 3.3. ^13^C NMR

The ^13^C NMR analysis method used was based on a previously reported study [7,10,11,14], which reported a carbonyl peak at δ 175.0 and characteristic five aliphatic carbon signals at δ 34.2, 31.2, 24.6, 22.3, and 13.9. Lipid (180 mg) extracted from the salmon roe samples was dissolved in 700 μL of deuterated chloroform (CDCl_3_; Cambridge Isotope Laboratories, Inc., USA) and then transferred to a 5 mm NMR tube. The ^13^C NMRs (100 MHz) were conducted using a Varian 400 MHz VNMRS instrument (Varian, Inc., Pullman, WA, USA) using a 5 mm Oneprobe^TM^, and the running conditions used were those as described previously [7,14,42]. Chemical shifts for both carbon and proton were referenced to a region peak of CDCl_3_ at 77.06 ppm. The NMR spectra were analyzed using MestReNova version 12.0 software (Mestrelab Research, Bajo, Santiago de Compostela, Spain). Peak assignment identification was carried out according to previously reported literature [43]. Area percentage of fatty acids was obtained from the area integrator response of the NMR spectra (MestReNova version 12.0 software, Mestrelab Research, Bajo, Santiago de Compostela, Spain). 

### 3.4. Statistical Analysis

Measurements of three individual replicates of fresh salmon roe, salted roe, and fermented roe samples were conducted in triplicate and data were reported as mean ± standard deviation. Statistical analysis of data was performed using Minitab^®^ Software (Version 16.0, Minitab Inc., Sydney, NSW, Australia). The homogeneity of variance and normality of the data were determined using Bartlett’s test and Shapiro–Wilk test, respectively. General linear model was carried out to determine analysis of variance (ANOVA), and Tukey’s test was conducted to separate the means and to determine the effects of processing treatments on fatty acid distributions in processed salmon samples at *p* < 0.05. The model used was Yij = μ + τi + ϵij, where μ is the experiment common effect, τ the treatment effects, and ϵ is the random error.

## 4. Conclusions

In the present study, ^13^C NMR was used to evaluate the effect of salting and fermentation on the positional distribution of fatty acids in salted and fermented Chinook salmon roe. ^13^C NMR enabled comparative analysis of the positional distribution of fatty acids in the lipids present in salted and fermented Chinook salmon roe samples, based on five spectral regions (carbonyl region, double bond/olefinic, glycerol backbone, aliphatic group, and chain ending methyl group). From the results obtained, substantial differences were found among the regiospecific distribution of major omega-3 fatty acids (EPA and DHA). DHA was the main esterified fatty acid found at the *sn*-2 position, while other fatty acids, such as EPA and SDA, were found to be randomly distributed or preferentially esterified at the *sn*-1 and *sn*-3 positions. It is of interest that a higher DHA content at the *sn*-2 position was found compared that at the *sn*-1,3 position of TAGs in lipid extracted from fermented salmon roe samples. DHA esterified at the *sn*-2 position is reported to be highly bioavailable and able to cross the blood/brain barrier, indicating that consumption of fermented salmon roe could facilitate brain and mental health. Hence, it is anticipated that the results of the present study would be of interest for the production of processed marine fish sourced material, such as roe, that has an enhanced proportion of DHA or EPA at the *sn*-2 position in TAGs.

## Figures and Tables

**Figure 1 molecules-28-00454-f001:**
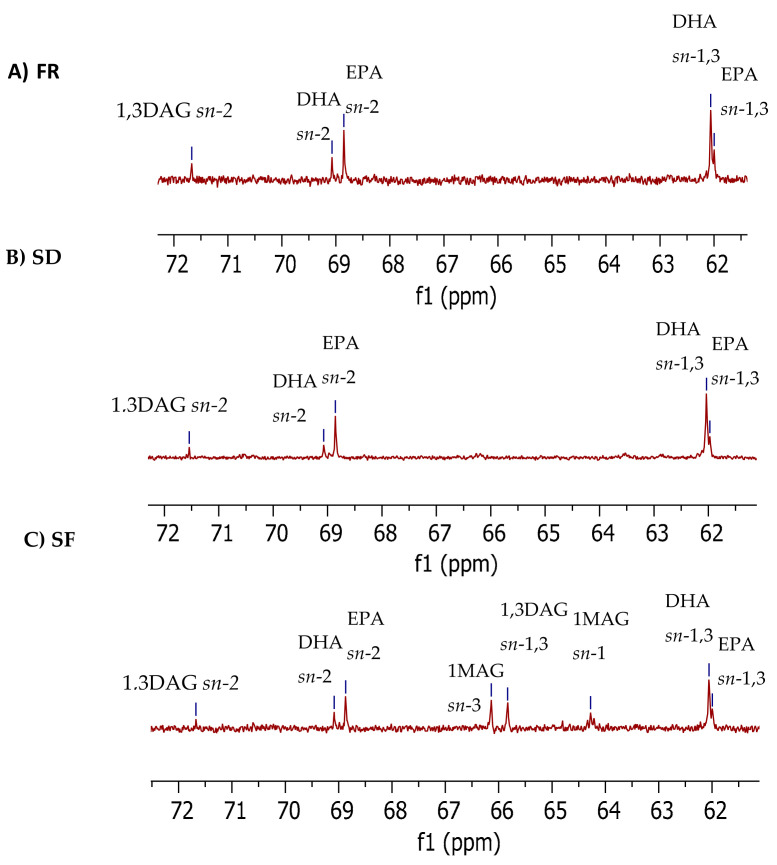
Glycerol carbon region of ^13^C NMR spectra of fresh (**A**), salted (**B**), and fermented salmon (**C**) roe lipids. Chemical shift for glycerol carbon region is 72.0–61.0 ppm. Abbreviations: MAG = monoacylglycerol, DAG = di-acylglycerol, EPA = eicosapentaenoic acid, DHA = docosahexaenoic acid. FR= raw or fresh salmon roe; SD = salted dried salmon roe; SF = fermented salmon roe.

**Table 1 molecules-28-00454-t001:** ^13^C NMR carbonyl region fatty acid *sn*-2 and *sn*-1,3 positional distribution in lipid extracted from fresh (FR), salted (SD), and fermented (SF) salmon roe and their location.

Fatty Acids	δ (ppm)		FR Roe Lipids Integrated Area (%)		SD Roe Lipids Integrated Area (%)		SF Roe Lipids Integrated Area (%)	
	*sn*-2	*sn*-1,3	*sn*-2	*sn*-1,3	*sn*-2	*sn*-1,3	*sn*-2	*sn*-1,3
22:6n-3 (DHA)	172.08	172.27	6.8 ± 0.4 ^B^	1.0 ± 0.1 ^A^	7.1 ± 1.2 ^B^	0.8 ± 0.1 ^AB^	10.1 ± 1.1 ^A^	0.7 ± 0.1 ^B^
22:5n-3 (EPA)	172.37	172.77	1.3 ± 0.23 ^A^	14.7 ± 1.5	0.4 ± 0.1 ^B^	18.5 ± 2.8	0.7 ± 0.1 ^B^	15.7 ± 0.8
C18:4n-3 (SDA)	172.39	172.92	0.9 ± 0.1 ^A^	2.7 ± 0.3 ^B^	0.6 ± 0.1 ^B^	4.6 ± 1.3 ^AB^	1.1 ± 0.1 ^A^	4.9 ± 0.7 ^A^
SFA	172.48	173.20	2.8 ± 0.2	25.3 ± 1.4 ^A^	3.0 ± 1.3	15.7 ± 1.6 ^B^	3.0 ± 0.3	19.1 ± 1.6 ^B^
MUFA	174.46	173.18	7.2 ± 1.1 ^A^	37.3 ± 1.0	4.5 ± 1.2 ^B^	44.9 ± 5.7	5.7 ± 0.3 ^AB^	39.2 ± 1.4

^A,B^ Values with different superscripts within a row are significantly different (*p* < 0.05). Abbreviations: SFA, saturated fatty acids; MUFA, monounsaturated fatty acids; SDA, stearidonic acid; EPA, eicosapentaenoic acid; DHA, docosahexaenoic acid. The structure of the n-3 fatty acids (SDA, EPA, and DHA) is provided in Appendix A.

**Table 2 molecules-28-00454-t002:** ^13^C NMR olefinic region of lipids extracted from fresh (FR), salted (SD), and fermented (SF) salmon roe.

Assignment	δ	FR Roe Lipids	SD Roe Lipids	SF Roe Lipids
	(ppm)	Integrated Area %	Integrated Area %	Integrated Area %
C4, n-3	126.98	7.0 ± 0.2 ^A^	5.9 ± 0.4 ^B^	4.0 ± 0.01 ^C^
C4, 22:6n-3	127.59	4.1 ± 0.2 ^B^	7.8 ± 0.7 ^A^	2.5 ± 0.2 ^C^
22:5n-3 20:5n-3 20:4n-3 18:4n-3	128.03	17.2 ± 0.6 ^B^	20.0 ± 0.4 ^A^	9.2 ± 0.2 ^C^
C6, 20:5n-3 20:4n6	129.52	17.5 ± 0.7 ^A^	6.7 ± 0.4 ^B^	5.3 ± 0.2 ^C^
C5, 20:5n-3 20:4n6	128.78	1.1 ± 0.2 ^B^	2.2 ± 0.03 ^B^	9.1 ± 0.1 ^A^
C5, 22:6n-3	129.42	3.7 ± 0.2 ^B^	5.5 ± 0.4 ^A^	2.6 ± 0.02 ^C^
16:1	129.77	1.9 ± 0.1 ^B^	3.9 ± 0.3 ^A^	1.6 ± 0.1 ^B^
18:1	129.66	10.1 ± 0.4 ^B^	12.8 ± 0.5 ^A^	5.9 ± 0.1 ^C^
20:1	129.96	14.5 ± 3.0 ^AB^	17.6 ± 1.5 ^A^	10.0 ± 0.5 ^B^
22:1	129.87	3.7 ± 0.2 ^B^	4.4 ± 0.13 ^A^	1.6 ± 0.3 ^C^
18:2n6	130.05	1.7 ± 0.1 ^C^	5.0 ± 0.1 ^A^	2.0 ± 0.1 ^B^
18:2n-3	130.30	4.0 ± 1.08	4.9 ± 1.7	4.2 ± 1.4
20:4n6	130.42	2.2 ± 0.3 ^A^	1.3 ± 0.2 ^B^	0.9 ± 0. 1 ^B^
β-eleostearic acid	130.84	ND	ND	17.9 ± 0.9
C3, n-3	131.96	11.1 ± 1.2 ^A^	5.9 ± 0.2 ^B^	4.3 ± 0.4 ^B^

^A–C^ Values with different superscripts within a row are significantly different (*p* < 0.05). Abbreviation: C = carbon, n-3 = omega-3.

**Table 3 molecules-28-00454-t003:** ^13^C NMR glycerol carbon region of lipids extracted from fresh (FR), salted (SD), and fermented (SF) salmon roe.

Assignment	Position		FR Roe Lipids	SD Roe Lipids	SF Roe Lipids
		δ (ppm)	Integrated Area %	Integrated Area %	Integrated Area %
1,3-DAG	*sn*-2	71.60	6.0 ± 0.6 ^A^	6.3 ± 0.3 ^A^	2.3 ± 0.2 ^B^
	*sn*-1,3	64.80	ND	ND	5.5 ± 0.2
22:6n-3 (DHA)	*sn*-2	69.07	10.4 ± 0.4 ^A^	8. 5 ± 0.4 ^B^	6.3 ± 0.2 ^C^
	*sn*-1,3	62.05	46.6 ± 0.8	44.7 ± 1.3	44.4 ± 0.7
22:5n-3 (EPA)	*sn*-2	68.86	19.5 ± 0.6 ^B^	22.0 ± 0.7 ^A^	15.7 ± 1.4 ^C^
	*sn*-1,3	62.00	14.7 ± 1.5	18.5 ± 2.8	15.7 ± 0.8
1-MAG	*sn*-1	64.28	ND	ND	10.2 ± 0.5
3-MAG	*sn*-3	65.83	ND	ND	7.9 ± 0.4

Each value is the mean ± standard deviation of three replicate samples (*n* = 3). ^A–C^ Values in the same column followed by a different superscript are significantly different (*p* < 0.05). Abbreviations: MAG = monoacylglycerol, DAG = di-acylglycerol, EPA = eicosapentaenoic acid, DHA = docosahexaenoic acid.

**Table 4 molecules-28-00454-t004:** ^13^C NMR aliphatic and methyl end region of lipids extracted from fresh (FR), salted (SD), and fermented (SF) salmon roe.

Assignment	Chemical Shift	FR Roe Lipids	SD Roe Lipids	SF Roe Lipids
	δ (ppm)	Integrated Area %	Integrated Area %	Integrated Area %
(a). Aliphatic region				
C3, n-6	31.49	15.3 ± 0.9 ^B^	9.8 ± 0.6 ^C^	18.9 ± 0.8 ^A^
C3, n-7	31.75	9.6 ± 0.3	8.3 ± 0.7	9.0 ± 0.9
C3, n-9	31.88	49.7 ± 0.6 ^B^	56.0 ± 0.4 ^A^	44.7 ± 0.4 ^C^
SFA	31.90	25.4 ± 1.5	25.9 ± 0.5	27.5 ± 1.4
(b). Methyl end chain region				
C1	14.08	80.2 ± 0.8	81.2 ± 0.7	80.5 ± 0.8
C1, n-3	14.24	19.8 ± 0.8	18.8 ± 0.7	19.5 ± 0.8

Each value is the mean ± standard deviation of three replicates (*n* = 3). ^A–C^ Values in the same row followed by a different superscript are significantly different (*p* < 0.05). Abbreviations: FR = fresh salmon roe lipid extract; SD = salted salmon roe lipid extract; SF = fermented salmon roe lipid extract. Chemical shift regions were 72.0–61.0 ppm.

## Data Availability

Data are available upon request.

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
