# Peer review of "Positional Distribution of Fatty Acids in Processed Chinook Salmon Roe Lipids Determined by 13C Magnetic Resonance Spectroscopy (NMR)"

_molecules, 2023, doi:10.3390/molecules28010454_

Round 1

Reviewer 1 Report

It is my pleasure to review the manuscript submitted by Bunga et al. (molecules-2098874).

In this article, they mainly reported the positional distribution of fatty acids of Chinook salmon roe by applying 13C NMR spectral analysis. The authors have used the 13C NMR (100 MHz) spectrum of extracted sample/mixture for the study. Thus, it is somewhat difficult to assign signals for each n-3 acid from the mixture spectrum. Moreover, 13C NMR can be applied to study the positional distribution of different fatty acids on the glycerol backbone in a semi-quantitative way. It could be interesting/appropriate if the authors can – (i) separate the mixtures, (ii) scan 1H NMR, and (iii) conduct nuclear Overhauser effect (nOe) in addition to 13C NMR.

Some of the necessary corrections are mentioned below:

[1] Title: --

[2] Abstract and keywords:

- Several typos in this section must be improved. Please write 13C (not 13C) and sn-2 (not sn-2).

[3] Introduction:

- Please write ‘1H NMR, 13C NMR, and 31P NMR’   instead of ‘1H-NMR (proton), 13C-NMR (carbon), and 31P-NMR (phosphorus)’. Follow a similar style throughout the manuscript. 

[4] Materials and methods:

- Although authors mentioned ‘The 13C- and 1H-NMR experiments were conducted….’ , there are no 1H NMR spectra in the manuscript.

- If authors are using Varian 400 MHz VNMRS instrument (Varian, Inc., USA), 13C NMR must be 100 MHz. Please mention it properly for 13C NMR.

- In section 2.3, please add a reference with reference 14 for more information on NMR spectra (DOI: https://doi.org/10.1016/j.molstruc.2020.128821).

[5] Results: In this section, authors must improve the typos of all subsections (For example, ‘3.1.13. C-NMR analysis of salmon roe extracted lipids’ should be ‘3.1. 13C NMR analysis of salmon roe extracted lipids’).

- For readers’ interest and better understanding, it is better to include a Figure containing structures of the fatty acids in the studied Chinook salmon roe samples.

- Once more, is it possible to compare Figure 1 with the 13C NMR spectrum of pure DHA, EPA, etc?

- In Figure 1C, please write 1MAG properly.

- The term ‘trans’ should be in the italic form (line 207).

- Please check for 1MAG sn-1, 1MAG sn-3, and 1,3DAG sn-1,3 (the position of acyl group/groups are mentioned twice in each). In addition, it should be 1-MAG, 3-MAG, or 1,3-DAG).

[6] Conclusion: Please write 13C NMR properly.

Author Response

The authors would like to thank the Editor for handling our manuscript, and the Reviewers for their comments, which have helped us to improve the overall quality of the manuscript. All the changes made in the MS are highlighted in red colour.

Reviewer #1

It is my pleasure to review the manuscript submitted by Bunga et al. (molecules-2098874). In this article, they mainly reported the positional distribution of fatty acids of Chinook salmon roe by applying 13C NMR spectral analysis. The authors have used the 13C NMR (100 MHz) spectrum of extracted sample/mixture for the study. Thus, it is somewhat difficult to assign signals for each n-3 acid from the mixture spectrum. Moreover, 13C NMR can be applied to study the positional distribution of different fatty acids on the glycerol backbone in a semi-quantitative way. It could be interesting/appropriate if the authors can – (i) separate the mixtures, (ii) scan 1H NMR, and (iii) conduct nuclear Overhauser effect (nOe) in addition to 13C NMR.

Response:  We thank Reviewer for their feedback.  We performed 13C, and 1H NMR for our samples in three separate studies. The result obtained from 1H NMR has been used to determine fatty acid composition, but the results are not as good as the GC results reported in our earlier work [Bunga, S. J.; Ahmmed, M. K.; Lawley, B.; Carne, A.; Bekhit, A. E.D. A., Physicochemical, biochemical and microbiological changes of jeotgal-like fermented Chinook salmon (Oncorhynchus tshawytscha) roe. Food Chemistry 2023, 398, 133880; Bunga, S.; Ahmmed, M. K.; Carne, A.; Bekhit, A. E.-D. A., Effect of salted-drying on bioactive compounds and microbiological changes during the processing of karasumi-like Chinook salmon (Oncorhynchus tshawytscha) roe product. Food Chemistry 2021, 357, 129780]. Therefore, we feel it will not add to the overall story. 13C NMR spectral analysis would be enough for the purpose of this study.  Most of the studies have used only 13C NMR for the investigating positional distribution. Some links for your reference.

https://onlinelibrary.wiley.com/doi/abs/10.1002/ejlt.201300357

https://www.sciencedirect.com/science/article/pii/000930849502462R

https://aocs.onlinelibrary.wiley.com/doi/abs/10.1007/BF02534402

https://link.springer.com/article/10.1007/s11746-009-1370-y

https://link.springer.com/article/10.1007/s11746-010-1638-2

considering the separation of individual compounds is an extremely involving process that our lab is not set up to conduct such research.

Some of the necessary corrections are mentioned below:

[1] Title: --

Response: We checked the title and kept unchanged as no correction was suggested.

[2] Abstract and keywords:

- Several typos in this section must be improved. Please write 13C (not 13C) and sn-2 (not sn-2).

Response: Typos has been fixed

[3] Introduction:

- Please write ‘1H NMR, 13C NMR, and 31P NMR’   instead of ‘1H-NMR (proton), 13C-NMR (carbon), and 31P-NMR (phosphorus)’. Follow a similar style throughout the manuscript. 

Response: Corrected and followed the same style throughout the manuscript.

[4] Materials and methods:

- Although authors mentioned ‘The 13C- and 1H-NMR experiments were conducted….’ , there are no 1H NMR spectra in the manuscript.

Response: We thank Reviewer for the comment. We sincerely apologize for this unintentional typo. We have corrected this in the revised version.

- If authors are using Varian 400 MHz VNMRS instrument (Varian, Inc., USA), 13C NMR must be 100 MHz. Please mention it properly for 13C NMR.

Response: Mentioned properly

“The 13C-NMR (100 MHz) were conducted using a Varian 400 MHz VNMRS instrument (Varian, Inc., USA) using a 5 mm OneprobeTM, and the running conditions used were those as described previously [7, 14, 22].”

- In section 2.3, please add a reference with reference 14 for more information on NMR spectra (DOI: https://doi.org/10.1016/j.molstruc.2020.128821).

Response: We added three references 7, 10, 11 which is more relevant to the work than a research on mannopyranoside esters.

“The 13C-NMR analysis method used was based on a previously reported study [7, 10, 11, 14], which reported a carbonyl peak at δ 175.0 and characteristic five aliphatic carbon signals at δ 34.2, 31.2, 24.6, 22.3, and 13.9”

[5] Results: In this section, authors must improve the typos of all subsections (For example, ‘3.1.13. C-NMR analysis of salmon roe extracted lipids’ should be ‘3.1. 13C NMR analysis of salmon roe extracted lipids’).

Response: Done

- For readers’ interest and better understanding, it is better to include a Figure containing structures of the fatty acids in the studied Chinook salmon roe samples.

Response: We than Reviewer for the suggestion.  To meet the Reviewer comment, we have added structure on

Figure S4: Structure of major n-3 fatty acids.

- Once more, is it possible to compare Figure 1 with the 13C NMR spectrum of pure DHA, EPA, etc?

Response: In this study we did not purify the EPA and DHA from roe. So, comparing Figure 1 with the 13C NMR spectrum of pure DHA, EPA is out of the scope.

- In Figure 1C, please write 1MAG properly.

Response: Done

- The term ‘trans’ should be in the italic form (line 207).

Response: Done

- Please check for 1MAG sn-1, 1MAG sn-3, and 1,3DAG sn-1,3 (the position of acyl group/groups are mentioned twice in each). In addition, it should be 1-MAG, 3-MAG, or 1,3-DAG).

Response: We double checked the result and updated 1-MAG, 3-MAG, or 1,3-DAG as suggested by Reviewer,   1-MAG, 3-MAG, or 1,3-DAG.

[6] Conclusion: Please write 13C NMR properly.

Response: Done. Wrote 13C NMR instead of 13C-NMR

Reviewer 2 Report

Please see comments given in the text of reviewed attached file of manuscript.

Author Response

Reviewer #2

It is better to add a background of your research and importance of your research in 1 or 2 sentences.

Response: Done.

“Abstract: Recently, there has been great interest in the lipidomic of marine lipids and their potential health benefits. Processing of seafood products can potentially modify the characteristics and composition of lipids. The present study investigated the effect of processing methods (salting and fermentation) on the positional distribution of fatty acids of Chinook salmon roe using 13C nuclear magnetic resonance spectroscopy (NMR).”

Please add ANOVA table in the results section.

Response: The analysis of variance for each fatty acid/fatty acid group will result in large number of tables (20 for table 2 only). This is excessive. The aim from conducting the statistical analysis is to separate the means and identify where differences exist. This has been accomplished in the current format.

please add your used model and its components in the text of manuscript

Response: done. The following sentence has been added “The model used was Yij=μ+τi+ϵij, where μ is the experiment common effect, τ the treatment effects, and ϵ is the random error.”

Round 2

Reviewer 1 Report

It is my pleasure to review the revised manuscript submitted by Bunga et al. (molecules 2098874-v2).

In my review report, I have requested the authors address several points. The authors are unable to change/improve most of the points. They have just fixed some typos, grammatical errors, etc. Here are some more corrections:

(1) In line 51, 1H-NMR, 13C-NMR,’ need to be corrected.

(2) Lines 184, 185, 251, etc. need to be corrected.

(3) In Section 2.3, I have requested to add a reference as it is related to 13C NMR spectra (various aliphatic, carbonyl peak regions are mentioned there), which might be helpful for readers.

(4) A careful revision of the manuscript is essential.

Author Response

It is my pleasure to review the revised manuscript submitted by Bunga et al. (molecules 2098874-v2). In my review report, I have requested the authors address several points. The authors are unable to change/improve most of the points. They have just fixed some typos, grammatical errors, etc. Here are some more corrections:

Response: We thank Reviewer for the comment. All possible corrections have been done in the previous version; we provided justification where we were unable to do the change. The feedback from the Reviewer really helped to improve the quality of manuscript.

(1) In line 51, ‘1H-NMR, 13C-NMR,’ need to be corrected.

Response: Done

(2) Lines 184, 185, 251, etc. need to be corrected.

Response: Done

(3) In Section 2.3, I have requested to add a reference as it is related to 13C NMR spectra (various aliphatic, carbonyl peak regions are mentioned there), which might be helpful for readers.

Response: Done

(4) A careful revision of the manuscript is essential.

Response: Done